# Automating the Process of Energy-Technology Invention

**David E. Tew,** [1] **Mihaela D. Quirk,** [1] **Lipi R. Acharya** [1]

[1]Advanced Research Projects Agency-Energy, U.S. Department of Energy
1000 Independence Ave. SW
Washington, DC 20585
{David.Tew, Mihaela.Quirk}@hq.doe.gov

## Abstract

Artificial Intelligence (AI) is facilitating transformative productivity enhancements by enabling ever-increasing levels of automation. The Advanced Research Projects Agency-Energy (ARPA-E) of the U.S. Department of Energy seeks to leverage this capability to automate elements of the energy technology development process in order to accelerate the realization of technological solutions to our climate crisis. This paper introduces three critical AI-enabled design capabilities – expert optimizers, low-cost yet high-fidelity evaluators, and inverse design tools – that are being developed for a wide range of energy applications and provides examples of two such tools being developed to accelerate the discovery of new materials.

The transition to a zero-carbon energy system in a time frame that is commensurate with the $1.5°C$ goal set by the Intergovernmental Panel on Climate Change requires the rapid development of a wide range of energy technologies. However, to develop them in adequate time, transformative design methods with disruptive potential are required.

In order to accelerate the development of such tools, ARPA-E launched the Design Intelligence Fostering Formidable Energy Reduction and Enabling Novel Totally Impactful Advanced Technology Enhancements (*i.e.*, DIFFERENTIATE, or $D'$) program in April 2019, and through it is investing \$35M to develop a suite of design tools enhanced by AI and machine learning (ML). These include expert optimizers, low-cost yet high-fidelity design evaluators, and generative/inverse design models for range of energy applications, including thermodynamic systems, electrical circuits, molecules and materials, aerodynamic components, and photonic devices (ARPA-E 2019).

The process that engineers use to develop new technologies and design new products is largely founded upon, and is similar to, the scientific method. Engineers hypothesize solutions to a problem, evaluate them via simulations or experiments, and iterate as required to ideally achieve performance levels that are consistent with the problem definition. This simplified design process is depicted in Figure 1 and may be used as a framework to define the desired roles of productivity-augmenting ML-based design tools.

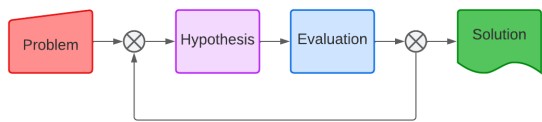

Figure 1: Engineering design process.

In the traditional, or forward, design process, cost is driven by the product of number of required iterations and the cost per iteration – with the cost per iteration being driven by the expense of evaluating candidate designs. Alternatively, if the inverse of the combined hypothesis and evaluation functions can be found, the design process itself may be inverted in that designs may be expressed as explicit functions of the desired output performance parameters, and these inverse representations may be subsequently used to rapidly generate design concepts from requirements.

## Desired Capabilities

Design tools are sought to enhance both the creativity of engineers as they develop new ideas, and the efficiency with which they evaluate them. Such tools might come in the form of domain-expert optimizers, high-fidelity and low-cost evaluators, or generative/inverse models.

Domain-expert optimizers may leverage ML methods in the development of designs in a specific domain (*e.g.*, thermodynamic systems, electrical circuits, or materials) using data from physics-based numerical simulations (*e.g.*, density functional theory, computational fluid dynamics) and/or physical experiments. These domain-expert tools can be more efficient than general-purpose optimizers (Yoon et al. 2021).

To reduce the cost of the data required to train expert optimizers, ML-based surrogate models may be developed and employed to evaluate the performance of the many candidate designs required for training purposes (Willcox, Ghattas, and Heimbach 2021). These models are frequently in the form of neural networks (NNs) or Gaussian processes (GPs), and they may be developed with data that was either previously acquired or generated during the optimization process (Cozad, Sahinidis, and Miller 2014; Rackauckas et al. 2021). Once trained, these models can offer effectively instanta-

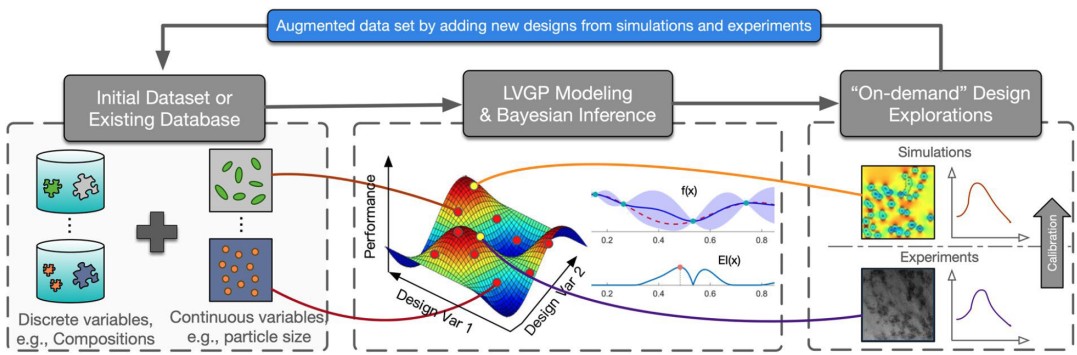

Figure 2: The Bayesian optimization framework for materials design with LVGP-based surrogate model. Figure reproduced from (Zhang, Apley, and Chen 2020); licensed under a Creative Commons Attribution (CC BY) license.

neous design performance predictions. In this context, ML surrogate models would offer an attractive value proposition if the number of current design evaluations required to train them is less than the number of new evaluations that would be required to train the expert optimizer without a surrogate model.

Fortunately, multiple strategies may be employed to reduce the time and/or cost required to develop such models. First, both low- and high-fidelity simulation data may be merged into meta-model representations that provide the flexibility for designers to trade accuracy versus cost. Secondly, a diverse set of desired simulations that span the expected input space may be preselected and run in parallel – potentially offering a robust source of training data in a short period of (clock) time, but perhaps at the cost of additional processing time. Lastly, some approaches offer predictions of nominal performance along with an estimate of uncertainty. This uncertainty information can then be used during the course of ML training or traditional optimizations to trigger the refinement of the surrogate model (Zhang, Apley, and Chen 2020).

Finally, to avoid the cost and time associated with design iteration, NNs may be used to express designs as explicit functions of their requirements, thus "inverting" the design process. Such a capability may take the form of invertible NNs that are trained in the forward direction per the above-described techniques, but may be used in practice in the reverse direction, where samples of designs may be generated using desired requirements (Ardizzone et al. 2018; Lee et al. 2021).

## Progress

Progress made thus far in the development of the three desired capabilities is described in this section and exemplified with two $D'$ projects that are seeking to accelerate the design of new materials for energy-related applications.

### Expert Optimizers

In the context of $D'$, expert optimizers are ML methods trained against optimality criteria in narrow application spaces. For example, optimizers based on reinforcement learning (RL) methods are being investigated in several

"system" architecture design applications. In RL, an agent interacts with its environment – the *world model* – to maximize a reward that expresses optimality criteria. In the context of energy technology design, relevant worlds include the design problem definition and an associated design performance estimation approach. A representative project for this optimality quest is led by Carnegie Melon University and aims at developing RL-based catalyst design tools (Yoon et al. 2021).

### Low-Cost High-Fidelity Function Evaluators

The cost-effective evaluation of a design candidate's performance is critical in both traditional and ML-enhanced optimization processes, as well as in the development of inverse design tools. Classical evaluations involve experiments or high-fidelity numerical simulations, whereas ML techniques lead to surrogate models that can efficiently combine both historical and current data from simulations and/or experiments.

There has been a significant focus on both the development and use of surrogate models as low-cost high-fidelity evaluators throughout the $D'$ program.

As an example, Northwestern University is developing an ML-based mixed-variable optimization framework for designing new materials, with a specific focus on those that exhibit metal-to-insulator transitions (MITs) that leverages surrogate models among other ML-based methods. The challenge characterizing the identification of these materials is fivefold: the high-dimensionality of the atomic structure-composition variable space, the prohibitive cost of high-fidelity simulations, the lack of unified literature reporting, the complexity of the physics, and a disjoint design space due to mixed qualitative and quantitative design variables.

The team employs several ML methods – including natural language processing (NLP), conditional variational autoencoders (CVAEs), and Bayesian optimization (BO) with latent-variable Gaussian processes (LVGPs), latent map Gaussian processes (LMGPs), and binary tree-based models – to discover new MITs materials. More specifically, NLP, CVAEs, active learning, and binary classification models are being developed and used to screen materials concepts to identify candidate MITs families, as well as synthesis paths

from literature data extracted from millions of journal articles and papers (Georgescu et al. 2019; Georgescu and Millis 2021; Jensen et al. 2019; Kim et al. 2020).

Unlike standard GPs that operate on numerical/quantitative data, LVGPs automatically discover the qualitative-to-quantitative latent variable mapping. Consequently, LVGPs provide a physics-based dimensionality reduction, while also lowering the number of simulations required for high accuracy (Zhang et al. 2019). Moreover, LMGPs extend GPs to accommodate qualitative inputs and handle categorical inputs of variable length, rendering a single latent space with insights of the underlying physics.

An overview of the BO framework with LVGP-based surrogate model is depicted in Figure 2 (Zhang, Apley, and Chen 2020). Another recent effort includes combining LVGP-based multi-objective BO with high-fidelity density functional theory-based simulations to optimize simultaneously the bandgap tunability and thermal stability in the lacunar spinels family of candidate MITs materials (Wang et al. 2020). In an exploration of less than $25\%$ of design space, this work has led to the identification of 12 new promising MITs spinel compounds. The team is collaborating with experimental research groups at other institutions (*e.g.*, UC Santa Barbara) to validate the newly identified materials, while continuing to search for new materials families.

This research targets future low-power microelectronic systems and it applies to other energy materials design and engineering problems with co-existing mixed variables and/or critical co-design of processing and structure.

### Generative/Inverse Design Tools

The inverse design approaches being developed within $D'$ are primarily based on NNs. These methods are being explicitly pursued for the design of photonic devices, aerodynamic surfaces, and heat exchangers. Techniques being employed include generative adversarial networks, normalizing flows, and variational encoders (Goodfellow et al. 2014). Deep generative models build new data, and even discover knowledge (Leoni et al. 2021) by revealing the distribution of the training set that matches that of greater ensembles in many physics-based applications (Wang and Wang 2021). Relatively recent work has suggested that ML-based material composition design tools can accelerate the identification of attractive new catalyst compositions (Gomez-Bombarelli et al. 2018).

Heterogeneous catalysts are broadly used in energy applications to facilitate the synthesis or the destruction of many chemical compounds by lowering the activation energies required for reactions to proceed, without being consumed. However, catalysts are often very expensive due to the use of platinum group metals.

New catalyst design efforts focus mostly on developing new compositions and/or surface morphologies that use less precious metals and/or have longer life. Multicomponent catalyst materials, such as perovskite oxides and spinels exhibit an electro-chemical stability that is superior to that of parent materials. However, the process of enhancing key properties of engineered catalysts suffers from a scarcity of

design principles and a multi-dimensional search space that is far too large to be investigated in its entirety.

The Massachusetts Institute of Technology (MIT) is leading a $D'$ project that combines expert optimizers and low-cost evaluations with inverse design to accelerate the development of catalysts that promote oxygen evolution and/or reduction reactions. The MIT team is tailoring non-platinum-group transition metal oxides to improve their catalytic performance and reduce the number of potential combinations required for testing. The ML approach employed successfully integrated synthesis data from the literature, simulations, lab-scale testing, and industrial prototyping to yield a catalyst design methodology that shows great promise for being faster and more efficient than traditional trial-and-error or serial experimentation-based approaches (Jensen et al. 2019; Kim et al. 2020).

The key ML-based techniques employed are deep generative models and message-passing NNs (Gilmer et al. 2017) for materials as well as convolutional NNs for machine vision to characterize catalysts. Figure 3 illustrates the property prediction strategy that leverages methods for generative modeling with auto-regressive representation (left). For each site, temperature, chemical potential, and previous sites within the same sample inform its subsequent choice of identity; that is, the probability of the fourth site $P_4$ depends on the previous sites $(P_1, P_2, P_3)$. On the right side, model and relative probabilities are shown, with the internal and chemical energies, and the entropy, in the training procedure to minimize the free energy of the model distribution (Damewood, Schwalbe-Koda, and Gomez-Bombarelli 2021; Gomez-Bombarelli et al. 2018). For certain perovskites, the Monte Carlo Tree Search algorithm with crystal graph convolutional NNs (Xie and Grossman 2018) was over 500 times more efficient in generating structures predicted to be stable. Under the auspices of $D'$, experiments based on more than 50 ML-predicted materials are being conducted at Argonne National Laboratory, and progress is being made in computer vision applications for spectroscopy data analysis at MIT.

## Current Challenges and Next Steps

Within $D'$ the above-described ML-enhanced design capabilities have indeed been demonstrated on multiple challenge problems (Ghosh et al. 2021; Quadir et al. 2021; Wu et al. 2021; Lee et al. 2021; Morehead et al. 2021; Anantharaman et al. 2021; Rackauckas et al. 2021). Once trained, the ML tools have been effective at optimizing materials/systems, rapidly evaluating design concepts, and even automatically generating designs from requirements. However, there are several outstanding challenges still being addressed. First, the cost of developing ML tools can be prohibitive in some applications, driving the need for strategies to reduce the number of data points required in training and/or the cost per point. Candidates include Bayesian optimization to identify and exploit high-value data regions, adaptive numerical methods that exploit optimally high cost/fidelity and low cost/fidelity model components at run time, and dimensionality reduction techniques that enable simpler NN representations with fewer weights/fitting

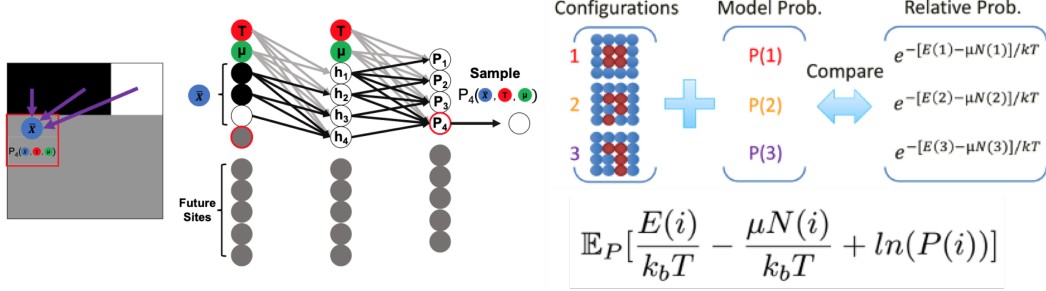

Figure 3: Autoregressive Generator: Leverage generative modeling methods to draw samples of the equilibrium distribution. Left: sampling using a neural network. Right: minimization of the free energy of the model distribution. Reproduced with permission and adapted from (Damewood, Schwalbe-Koda, and Gomez-Bombarelli 2021)

.

parameters, like graph NNs (Pandey et al. 2021).

Once an initial ML capability has been developed, transfer learning can improve initial guesses of network weights for new problems-yielding reduced training costs (Taylor and Stone 2009; Biagioni et al. 2020; St. John et al. 2019). As shown in the two examples, NLP methods can harvest training and validation data from literature, at a lower cost than that of running original experiments or physics-based simulations (Olivetti et al. 2020; Elton et al. 2019; Georgescu et al. 2021).

However, despite the multiple successful capability demonstrations within the program thus far, additional work is required to quantitatively understand the cost/benefit analysis afforded by ML-techniques in engineering design applications. Roughly speaking, the key question to be addressed is "Will enough designs be performed with the ML-tools to adequately amortize the cost of training?" Unfortunately, the answer to this question will undoubtedly depend on the application. Nevertheless, tools to estimate *a priori* the cost of developing an ML solution approach for a particular problem would be significant when balancing an investment in an AI/ML capability versus the use of traditional approaches. At present, the $D'$ project teams are continuing to move ahead with the maturation of their tools with a particular focus on increasing the complexity of their design problems through higher dimensionality due to increases in the fidelity of the design representation, the number of physical dimensions, and/or the physical phenomena considered (*e.g.*, fluid, structure, thermal, electrical, optical).

## Summary

As the climate crisis intensifies, our need for technological solutions that enable us to satisfy our energy needs while minimizing any associated climate impact will continue to increase. ARPA-E has launched the $D'$ program to develop ML-enhanced design tools that are intended to accelerate the development of energy-technology solutions to this challenge. Progress made thus far suggests that such tools will indeed have a very significant role to play. The three targeted design capabilities-expert optimizers, low-cost high-fidelity evaluators, and inverse/generative models have been demonstrated for initial challenge problems in multiple energy domains – including wind turbines, electrical circuits, heat transfer surfaces, catalysts, and semi-conductor materials.

Expert optimizers have been demonstrated to work in multiple domains; although, a full quantitative assessment of their performance versus traditional optimization approaches is still pending in most cases. Both GPs- and DNN-based surrogate models have been developed for multiple applications, and once trained, they have been shown to provide effectively instantaneous performance predictions of adequate fidelity within the bounds of the domains in which they were trained. Lastly, NN-based inverse, or generative models, that automatically generate designs given requirements appear to have a truly transformative potential, in part through their ability to dramatically lower the cost of bespoke designs.

## Acknowledgments

The authors gratefully acknowledge the support of ARPA-E for this work. However, the opinions expressed are those of the authors and are not necessarily those of ARPA-E or the U.S. Department of Energy.

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
