# OpenReview forum: "Automating the Process of Energy-Technology Invention"
_AAAI.org/2022/Workshop/ADAM — AAAI 2022 Workshop ADAM_

### Official Review · Reviewer_9Zt6 · 2021-11-30
**I give accept to this paper. Provides a good survey on design problems leveraging ML methodologies.**

**Rating:** 7
**Confidence:** 2

**Review:**

This paper provides a survey on design problems in terms of expert optimizers, low-cost high-fidelity evaluators, and inverse design tools. The author states that while the RL approach is still par with the traditional methodology, the authors' view in leveraging the surrogate models (GP and DNN) and generative models seems to have the potential to lower the cost of design problems.

I believe this is a solid survey paper on the current status of a design problem and fits well to the workshop.

---

### Official Review · Reviewer_Em9U · 2021-12-01
**Nice Meta-paper the describes the landscape of AI for Design. Will be even better if it can flesh out opportunities and open problems**

**Rating:** 7
**Confidence:** 4

**Review:**

This is a good survey paper based on performers of the DIFFERENTIATE initiative of ARPA-E.
The paper creates a taxonomy of AI approaches used in simulation-based design. This taxonomy consists of three classes - domain-expert optimizers, high-fidelity, and low-cost evaluators, and generative/inverse models.
This fits well to the workshop goals and will be a useful paper for the attendees.
Some suggestions:
1) The taxonomy is incomplete. There are other AI approaches in Design that may be worth noting to flesh out the taxonomy (perhaps these are not utilized by the D' performers and hence not included?)
2) It will be useful to lay out some of the open questions/opportunities in the field of Ai-enabled design (generalizability guarentees, sample complexity) and also make the case for open-source datasets.